# Chinese Residents’ Healthy Eating Intentions and Behaviors: Based on an Extended Health Belief Model

**DOI:** 10.3390/ijerph19159037

**Published:** 2022-07-25

**Authors:** Yiqin Wang, Xiaowei Wen, Ying Zhu, Yanling Xiong, Xuefan Liu

**Affiliations:** 1College of Economics & Management, South China Agricultural University, Guangzhou 510642, China; wyq1161012@stu.scau.edu.cn (Y.W.); zhuyingsom@stu.scau.edu.cn (Y.Z.); xylelin@stu.scau.edu.cn (Y.X.); xfliu@stu.scau.edu.cn (X.L.); 2Research Center for Green Development of Agriculture, South China Agricultural University, Guangzhou 510642, China

**Keywords:** health consciousness, health belief model, healthy eating intentions and behaviors, partial least squares structural equation modeling

## Abstract

Unhealthy eating is one cause of obesity and some chronic non-communicable diseases. This study introduces self-efficacy and health consciousness to construct an extended health belief model (HBM) to examine the factors influencing healthy eating intentions and behaviors of Chinese residents and explore the moderating effect of perceived barriers and the mediating effect of healthy eating intentions. Through the survey platform “Questionnaire Star”, this study collected quantitative data from 1281 adults, and partial least squares structural equation modeling was used for confirmatory factor analysis, path analysis, importance-performance map analysis, and multi-group analysis. Results showed that perceived susceptibility, perceived severity, perceived benefits, self-efficacy, and health consciousness had a significant positive effect on residents’ healthy eating intentions. Perceived barriers had a significant negative effect on residents’ healthy eating intentions. Healthy eating intentions had a significant positive effect on healthy eating behaviors. Perceived barriers played a significant negative moderating effect between healthy eating intentions and behaviors. Healthy eating intentions had a positive and significant mediating effect. The multi-group analysis showed that extended HBM has relative generalization ability. The extended HBM has good explanatory and predictive power for healthy diet and provides a new framework for understanding the influencing factors of individuals’ healthy eating intentions and behaviors.

## 1. Introduction

Changes in structure are one of the critical features of social transformation, and diet structure is directly related to human health [1]. After over 40 years of reform and opening up, China’s dietary structure and eating behavior have changed dramatically. Diet structure has shifted from the consumption of primarily grains and vegetables to more high-protein, high-fat foods [1]. Consequently, obesity and a series of chronic non-communicable diseases are increasing in China [2]. According to data released in the “Report on Nutrition and Chronic Diseases of the Chinese Residents (2020)”, the proportion of overweight or obese adult residents in China has reached 50.7% (Source: Official Website of the Information Office of the State Council of China. http://www.scio.gov.cn/xwfbh/xwbfbh/wqfbh/42311/44583/wz44585/Document/1695276/1695276.htm, accessed on 25 December 2020). Some studies have shown that China has the highest mortality rate from diet-related cardiovascular diseases (57.99%) among the 20 most populous countries in the world [2].

Empirical studies have shown that dietary intake is strongly associated with health, and low-quality diet and unbalanced energy intake are some of the important causes of health risks [3]. In the context of the general national overnutrition, a healthy diet is a diet consisting of vegetables, fruits, and whole grains, and one which contains less foods with high fat and high sugar (such as pies, cakes, and pastries) [4]. Diet is a changeable behavior, relatively speaking. Under-nutrition is often limited by objective factors such as the income level of the residents, while overnutrition depends more on the subjective dietary choices of individuals. Therefore, the diet associated with overnutrition is a more easily changeable behavior.

Understanding the determinants of eating behavior to improve an individual’s eating behavior is an important research area. Bettina et al. [5] concluded that patients with type 2 diabetes had poorer self-perception of their dietary healthiness and less intention of eating healthily, compared with patients with type 1 diabetes and the general population. Zhou et al. [6] believed that the factors affecting healthy dietary behavior include physiological factors, psychological factors, and social and cultural factors. Ji et al. [7] pointed out that the high educational level of the mother and the high annual family income are the protective factors of children’s healthy dietary behavior. Shi et al. [8] took middle school students as the research object and found that gender, education level, and dining place are important factors leading to students’ bad dietary behavior. Bouwman et al. [9] concluded that a brief self-intervention can promote healthy eating in a randomized intervention trial with Dutch residents. Alexandria et al. [10] applied the theory of planned behavior to examine healthy eating intentions and behaviors among African Americans and found that healthy eating intentions were a major predictor of eating behaviors. Chansukree and Rungjindarat [11] explored the influence of social cognitive determinants on healthy eating behaviors among Thai adolescents and showed that healthy eating among male adolescents was best predicted by perceived barriers.

Several scholars have applied a variety of behavioral theories to the study of health-related behaviors, such as the theory of rational behavior, the health belief model (HBM), and the theory of planned behavior. These theories have facilitated the understanding of the decision-making process of individuals’ health-related behaviors. Among them, HBM is one of the most influential theories for explaining and predicting individual health-related behaviors. It is commonly used in explaining health-related behaviors [12]. For example, Wang et al. [13] explored the relationship between health beliefs and non-communicable disease prevention behaviors and found that perceived barriers and self-efficacy had the greatest impact on behaviors. Kavanaugh [14] applied HBM to the food handling behavior of older adults and found that the construct of perceived susceptibility could be expanded. Chu and Liu [15] used HBM to predict the intention of Americans to receive the new crown pneumonia vaccine and showed that perceived benefits were positively associated with vaccination intention.

Many studies have focused on the healthy eating behaviors of residents in developed countries from the perspective of overnutrition; however, few studies have focused on healthy eating behaviors of residents in developing countries. Furthermore, there is limited literature on the application of HBM to study residents’ healthy eating behaviors, and there is no report to expand HBM on Chinese residents’ healthy eating behaviors. Consequently, this study proposes theoretical hypotheses based on the extended HBM and uses data from a sample of 1281 residents surveyed in China to study the factors influencing residents’ healthy eating intentions and behaviors by using partial least squares structural equation modeling (PLS-SEM). In addition, it examines the moderating effect of perceived barriers and the mediating effect of healthy eating intentions in residents’ decision-making on healthy eating behavior. The primary contributions of this paper are (1) to add self-efficacy and health consciousness to the traditional HBM and to verify the validity of this extended HBM in residents’ healthy eating decision-making; (2) to elucidate the key factors influencing Chinese residents’ healthy eating decisions.

## 2. Theoretical Framework and Research Hypothesis

The HBM is a common social cognitive model in health behavior research and contains four main components: perceived susceptibility, perceived severity, perceived benefits, and perceived barriers [16]. With later development, the model adds two cognitive constructs to enhance explanatory power independent of traditional HBM, namely self-efficacy [17] and health consciousness [18]. Based on this, we constructed the following research framework.

### 2.1. Perceived Susceptibility and Perceived Severity

Perceived susceptibility refers to a person’s perception of a health problem of infection [19] and is a major threat factor for food intake behaviors [20]. Perceived severity is a person’s judgment of the severity of the consequences of the issue [21] and has a direct effect on eating intentions and behaviors [22]. When individuals believe that a certain dietary behavior predisposes them to disease, they may develop a willingness to modify their diet. Therefore, we propose the following hypotheses.

**H1.** 
*Perceived susceptibility has a significant positive effect on residents’ healthy eating intentions.*


**H2.** 
*Perceived severity has a significant positive effect on residents’ healthy eating intentions.*


### 2.2. Perceived Benefits and Perceived Barriers

Perceived benefits and perceived barriers cooperate to determine the occurrence of a specified action. In some situations, perceived barriers offset some of the perceived benefits, while in other situations, the opposite is true [23]. In this study, perceived benefits refer to an individual’s beliefs regarding the relative effectiveness of an action to reduce the disease threat [24]. Individuals act based on their awareness of certain benefits. Perceived barriers refer to the inconvenience or unattractiveness of a certain behavior to the individual and it prevents the individual from adopting the behavior [24]. In addition, perceived barriers have been shown to be the most significant influence on intention in HBM [21]. Therefore, we propose the following hypotheses.

**H3.** 
*Perceived benefits have a significant positive effect on residents’ healthy eating intentions.*


**H4.** 
*Perceived barriers have a significant negative effect on residents’ healthy eating intentions.*


### 2.3. Self-Efficacy

Self-efficacy refers to the level of difficulty involved in performing a behavior. Self-efficacy is an important factor in peoples’ decisions to improve their health behaviors [25]. The theory of planned behavior establishes the influence of self-efficacy on intention. If a person has high self-efficacy to adopt a healthy diet, the person will have a stronger intention to implement these behaviors, and vice versa [26]. Studies have shown that self-efficacy can significantly affect eating intentions [22]. Therefore, we propose the following hypothesis.

**H5.** 
*Self-efficacy has a significant positive effect on residents’ healthy eating intentions.*


### 2.4. Health Consciousness

Health consciousness reflects the level of awareness of an individual’s health status and their intention to become healthier [27]. Health consciousness is considered a predictor of health attitudes and behaviors and is a direct determinant of health-related behavioral intentions [21,28]. People who are more health consciousness are more concerned about their health status and adopt health behaviors to prevent disease [29]. Therefore, we propose the following hypothesis.

**H6.** 
*Health consciousness has a significant positive effect on residents’ healthy eating intentions.*


### 2.5. Healthy Eating Intentions

Intentions are assumed to capture motivational factors that affect behaviors and have a strong positive correlation with behaviors [30]. A significant amount of evidence has shown the convergence between behavior intentions and actual behaviors [31,32]. A meta-analysis has shown that a medium-to-large change in intention leads to a small-to-medium change in behavior [33]. Furthermore, some studies have shown that eating intentions lead to eating behaviors [10]. Therefore, we propose the following hypothesis.

**H7.** 
*The healthy eating intentions of residents have a significant positive effect on their healthy eating behaviors.*


### 2.6. Moderating Effect of Perceived Barriers

Perceived barriers reveal beliefs that healthy eating is difficult to achieve, limiting adoption of healthy eating. This significantly reduces individuals’ intention to choose healthy foods [34]. Although higher intention leads to an increased willingness to adopt healthy eating behaviors, perceived barriers reduce the occurrence of behaviors, i.e., perceived barriers moderate the relationship between intention and behavior [30]. Therefore, we propose the following hypothesis.

**H8.** 
*Perceived barriers play a significant negative moderating role between residents’ healthy eating intentions and healthy eating behaviors.*


### 2.7. Mediating Effect of the Healthy Eating Intentions

In rational behavior theory and technology acceptance models, intention is considered the central factor linking internal cognitive beliefs to actual behavior [35]. Existing theoretical and empirical studies have shown that intention as a mediator affects the relationship between other variables and actual behavior [30]. Therefore, we propose the following hypothesis.

**H9a–f.** 
*Residents’ healthy eating intentions mediate the pathway of influence from perceived susceptibility, perceived severity, perceived benefits, perceived barriers, self-efficacy, and health consciousness to healthy eating behaviors, respectively.*


All associations hypothesized and tested associations are presented in Figure 1.

## 3. Data Source

### 3.1. Data Collection and Study Sample Design

Due to the COVID-19 pandemic, this study used China’s online survey platform “Questionnaire Star” for online research. Compared to field surveys, participants can provide more objective responses without interference [36]. We paid CNY 5 (about USD 0.7) (since this online survey took about 7 minutes to complete, a payment of CNY 5 (approximately USD 0.7) could make up for participant’s lost time and avoid the portfolio effect by paying them too much), through the survey platform “Questionnaire Star”, to every respondent who filled out one questionnaire. At the top of the questionnaire, the definition of related concepts such as healthy eating was provided so that the participants could understand before filling out the questionnaire. The survey was administered in December 2021. To qualify, individuals were required to be over 18 years of age. Individuals who could not promise to provide truthful answers, failed to check questions, or completed the survey out of the 95% confidence interval of survey time were excluded from our analysis. To ensure that each participant took the survey only once, the internet protocol address of each participant was tracked and checked. On average, the participants took 7 min to complete the survey, and 95% of the sample finished it in <15 min. A nationwide sample of 1787 respondents was collected, 506 invalid questionnaires were deleted, 1281 valid questionnaires were finally obtained (Participants in our sample were from 21 Chinese provinces, 3 Autonomous Regions, and 4 Municipalities directly under the Central Government, spreading across the northern and southern regions of China, with 57.6% in the north and 42.4% in the south), and the effective recovery rate of questionnaires was 71.68%.

Table 1 describes the sample which consisted of 710 females and 571 males. This aligns with the reality that Chinese women are the primary bearers of the family diet. The majority of respondents live in urban areas (72.13%) and a minority live in rural areas (27.87%). This is generally consistent with the seventh census of China (Source: China Statistical Yearbook of 2021. https://data.cnki.net/yearbook/Single/N2021110004, accessed on 1 January 2022). Compared to the population statistics, our sample is generally younger and well-educated. This is expected, given that the study recruited adult participants and was conducted online; it is reasonable to expect internet users to be better educated.

### 3.2. Survey Instrument

Each questionnaire contained three sections: the first section elaborated the purpose of the research, the research institution, and confidentiality. The second section investigated the socio-demographic characteristics of the respondents. The third section measured the structure of the HBM. A 5-point Likert-type scale was used for part three, with responses ranging from “strongly disagree” (1) to “strongly agree” (5). The content of the question items and the corresponding literature basis are shown in Appendix A. A forward–backward translation process was employed to ensure the content validity of the scales’ translation to Chinese. In addition, we invited relevant experts to pilot the questionnaire and provide feedback. Minor adjustments were applied based on their input.

## 4. Result and Discussion

The data were analyzed using SPSS 26.0 (Norman H. Nie, C. Hadlai Hull, and Dale H. Bent, Chicago, IL, USA) and Smart PLS 3.0 (Christian Ringle, Dipl.-Wilnf.Sven Wende, and Jan-Michael Becker, Oststeinbek, Germany). The relationships between variables were examined using PLS-SEM. This is a variance-based multivariate analysis tool for measuring path models with latent variables. PLS-SEM can avoid two serious problems: unacceptable solutions and factorial uncertainty.

### 4.1. Common Method Variance

Given that this study used self-reported methods to obtain the cross-sectional data, there may be issues of common method variance (CMV). Therefore, statistical analyses were conducted to examine the effect of CMV on the structure of the study. First, the 28 question items were tested using Harman’s single factor test through SPSS 26.0. The results show that the percentage of variance explained by the first common factor is 34.275%. This is less than the recommended value of 40%, which indicates that there is no serious problem of CMV (see Appendix B). Secondly, Smart PLS 3.0 was used for CMV tests [37]. The results show that the mean substantive factor loading for each indicator is 0.711, while the mean common method factor loading is 0.029, with a ratio of 24.5:1 (see Appendix C). This further established that common method bias is not a serious problem.

### 4.2. Reliability and Validity Analysis

The reliability and validity were analyzed using SPSS 26.0 and Smart PLS 3.0. Initially, 640 samples were randomly selected from 1281 samples for exploratory factor analysis. Subsequently, a confirmatory factor analysis was used to analyze the reliability and validity of the constructs for the remaining 641 samples. The results indicate that Bartlett’s test of sphericity is significant (*p* < 0.001) and that the Kaiser–Meyer–Olkin sampling adequacy measure is 0.930 > 0.8 (see Appendix C). This indicates that the data are appropriate for factor analysis. The 8 dimensions extracted are consistent with the hypothesized structure and are consistent with the 8 dimensions explored when the eigenvalue is greater than 1 (see Appendix B).

#### 4.2.1. Reliability Analysis

The data in Table 2 show that the Cronbach’s alpha for all the constructs ranges from 0.730 to 0.897 and is greater than 0.7. This indicates good internal consistency among the constructs [38].

#### 4.2.2. Validity Analysis

As shown in Table 2, the average variance extracted (AVE) of all the constructs ranges from 0.547 to 0.830, which is greater than the recommended value of 0.5 [39]. The composite reliability (CR) values range from 0.846 to 0.936—all greater than the recommended value of 0.7 [38]. The standardized factor loadings range from 0.718 to 0.928—all greater than 0.7 [40]. Thus, all the constructs have excellent convergent validity.

As shown in Table 3, the square root of the AVE of all constructs is greater than the Pearson correlation coefficient of that construct with other constructs [39]. In addition, the HTMT values are less than 0.85 [30]. Therefore, the constructs have good discriminant validity.

As shown in Table 2, the variance inflation factors (VIF) of all the constructs range from 1.000 to 2.452, which are less than the baseline value of 3.3 [41]. As shown in Table 3, the Pearson correlation coefficients are less than 0.7. Therefore, the covariance problem of the constructs in the structural model is not serious and does not affect the results.

### 4.3. Explanatory Power of Model

The explanatory power of the model was evaluated based on the PLS algorithm and blindfolding algorithm. From Table 4, the *R*^2^ values for healthy eating intentions and healthy eating behaviors are 0.563 and 0.358, respectively. This indicates that the model has moderately high explanatory power of the internal factor construct [40]. The explanatory effects *f*^2^ of the external factor construct on healthy eating intentions range from 0.151 to 0.212, which is a moderate effect explanatory power. The explanatory effect of healthy eating intentions on healthy eating behaviors with *f*^2^ of 0.351 is a high effect explanatory power [40]. In summary, the external factor construct has a moderate to high effect explanatory power on the internal factor construct. Based on the blindfolding program, the *Q*^2^ values of healthy eating intentions and healthy eating behaviors are 0.419 and 0.237, respectively. These are all greater than 0, indicating that the model has predictive relevance to the constructs [42]. In summary, the theoretical model has good explanatory and predictive power.

### 4.4. Path Analysis

This paper used the bootstrapping sampling 5000 times to test the model path. The results are displayed in Table 4 and Figure 2. The pathway results indicate that perceived susceptibility has a significant positive effect on healthy eating intentions, and H1 is supported. This result is consistent with existing studies [43]. This suggests that individuals with high perceived susceptibility will be more willing to adopt a healthy diet. The effect of perceived severity on healthy eating intentions is positive and significant and H2 is supported, which suggests that individuals’ beliefs about the severity created by unhealthy eating can explain healthy eating intentions. This aligns with the results of Diddana [44]. However, perceived severity (*β* = 0.093) has a lower effect on healthy eating intentions than perceived susceptibility (*β* = 0.110), and perceived severity has the smallest effect among the variables that positively predict healthy eating intentions.

Perceived benefits have a positive and significant effect on healthy eating intentions. H3 is established. This result aligns with previous studies [45]. The higher the perceived benefit, the higher the individual’s willingness to eat healthily. This suggests that educational materials must be emphasized to increase perceived benefits when formulating relevant healthy diet education projects. Perceived barriers have a negative significant effect on healthy eating intentions, and H4 is established, which indicates that perceived barriers prevent individuals from engaging in healthy eating. This aligns with previous research findings [46]. Furthermore, perceived benefits (*β* = 0.255) have a higher impact than perceived barriers (*β* = −0.045). This is contrary to the results of Janz and Becker [19], who concluded that perceived barriers are the most important predictor. This suggests that healthy eating requires individuals to establish new eating patterns, which can be a time-consuming and labor-intensive behavior, compared to an individual’s casual diet. Consequently, individuals have a high perception of the difficulty of performing this behavior.

The results of this study indicate that self-efficacy has a positive and significant effect on healthy eating intentions, and H5 is established. This result is similar to previous studies examining the effect of self-efficacy on behavioral intentions [47]. The degree to which an individual predicts the difficulty in engaging in a healthy diet will influence the likelihood of performing this behavior. Individuals who believe that eating healthy is easy or manageable will tend to eat healthy. Therefore, the improvement in individual self-efficacy during the implementation of a certain dietary behavior program should be emphasized.

Health consciousness has a significant positive effect on healthy eating intentions, and H6 is established. The finding of this study is supported by Versele et al. [28]. Individuals with higher health consciousness will increasingly focus on health-related information. Therefore, they are more capable of dealing with health-related issues and have a higher likelihood of implementing healthy eating behaviors. This subsequently leads to a greater intention to eat healthily. However, as shown in Table 4 and Table 5, the effect of health consciousness on healthy eating intentions is greater than the effect of health consciousness on healthy eating behaviors, and greater than the effect of health consciousness through healthy eating intentions to healthy eating behaviors (0.389 > 0.313 > 0.199). This highlights that raising health consciousness directly increases individuals’ intentions to eat healthily or their eating behavior. However, the effect of influencing behaviors through intentions is greatly diminished. Therefore, interventions to target health consciousness to diet should directly interfere with dietary intentions or dietary behaviors.

The impact of healthy eating intentions on healthy eating behaviors is positively significant, H7 is supported, which indicates that there is a convergence between healthy eating intentions and eating behaviors. This result is consistent with the study of Alexandria et al. that found that intention to eat healthy was a significant predictor of dietary intake behaviors [10]. Furthermore, perceived barriers play a significant negative moderating role between healthy eating intentions and behaviors. H8 is supported, which indicates that the slope of healthy eating intentions on behaviors decreases by 0.089 standard deviations when each unit of standard deviation of perceived barriers increases.

### 4.5. Mediation Analysis

The testing power of bootstrapping is higher than that of the Sobel test and can avoid errors in the mediation effect verification process [48]. Therefore, this paper used the bias-corrected nonparametric percentile bootstrap method to test for mediating effects on healthy eating intentions.

The findings suggest that healthy eating intentions play an important mediating effect in the proposed framework. This suggests that the new mechanism of HBM proposed in this paper is supported. As seen in Table 5, healthy eating intentions play a fully mediating role in perceived susceptibility (VAF = 0.947), perceived severity (VAF = 0.957), and perceived benefits (VAF = 0.949). This indicates that perceived susceptibility, perceived severity, and perceived benefits can only have an effect on individuals’ healthy eating behaviors through healthy eating intentions. Healthy eating intentions have a partial mediating effect on perceived barriers (VAF = 0.458), self-efficacy (VAF = 0.271), and health consciousness (VAF = 0.389). Interestingly, perceived susceptibility (0.248 > 0.014), perceived severity (0.269 > 0.012), and perceived benefits (0.300 > 0.017) indirectly influence healthy eating behaviors through healthy eating intentions stronger than their direct effects on behavior by themselves, and they do not have significant effects on healthy eating behaviors. Furthermore, perceived barriers (0.071 < 0.084), self-efficacy (0.148 < 0.398), and health consciousness (0.199 < 0.313) indirectly affect healthy eating behaviors through healthy eating intentions and are weaker than their direct effects on eating behaviors. Therefore, the direct impact of perceived barriers, self-efficacy, and health consciousness on healthy eating behaviors are higher than that through healthy eating intentions. Consequently, when implementing certain interventions for healthy eating based on HBM, for perceived susceptibility, perceived severity, and perceived benefits, dietary intention can be directly intervened. For perceived barriers, self-efficacy, and health consciousness, dual interventions on dietary intention and dietary behavior are recommended.

### 4.6. Multi-Group Analysis

Based on the social statistics characteristics of respondents, this study conducted a multi-group analysis to test whether the path model suitable for the entire sample is also suitable for specific groups. As shown in Table 6, gender, age, family income, BMI, marital status, residence, and employment status do not differ significantly by group. There was a slight significant difference in education between healthy eating intentions and healthy eating behaviors (*p* = 0.049); this suggests that increasing the education level of individuals can facilitate the conversion of intentions to behaviors. In general, the model framework constructed in this paper is generalizable and reproducible.

### 4.7. Importance Performance Matrix

This article used an IPMA to extend the PLS-SEM results by also taking the performance of each construct-measured on a scale from 0 to 100 into account. For a specific criterion construct, the IPMA contrasts the structural model total effects (importance) and the average values of the latent variable scores (performance) to highlight significant areas to improve management activities [49]. By combining the analysis of the importance and performance dimensions, the IPMA allows for prioritizing constructs to improve a certain target construct [50]. The specific results are shown in Table 7 and Figure 3.

Table 7 shows the outcomes of the IPMA, and it displays that health consciousness is the most vital factor in the performance of healthy dietary behaviors (0.258; 84.421), followed by perceived benefits (0.086; 83.998), perceived susceptibility (0.112, 83.706), self-efficacy (0.352, 74.046), perceived severity (0.085, 64.680), and perceived barriers (−0.030; 48.228). To better illustrate the IPMA results, we plotted Figure 3.

## 5. Conclusions and Suggestions

This paper explored the factors influencing healthy eating intentions and healthy eating behaviors using PLS-SEM. In addition, the moderating effect of perceived barriers and the mediating effect of healthy eating intentions were also tested. The following major conclusions and suggestions can be drawn from this research.

### 5.1. Conclusions

First, integrating self-efficacy and health consciousness into the HBM has good explanatory, predictive, and generalizing power in exploring dietary intentions and behaviors. Second, perceived susceptibility, perceived severity, perceived benefits, self-efficacy, and health consciousness are important factors that motivate individuals’ healthy eating intentions. However, perceived barriers inhibit individuals’ healthy eating intentions. Third, the order of effect is: health consciousness > perception benefit > perceived susceptibility > self-efficacy > perceived severity > perceived barriers. Fourth, healthy eating intentions have a positive effect on healthy eating behaviors, and perceived barriers negatively moderate the relationship between healthy eating intentions and healthy eating behaviors. Fifth, perceived susceptibility, perceived severity, perceived benefits, perceived barriers, self-efficacy, and health consciousness indirectly influence healthy eating behaviors through healthy eating intentions.

### 5.2. Suggestions

First, when designing and introducing influential dietary strategies, relevant departments should consider the positive effects of perceived susceptibility, perceived severity, perceived benefits, self-efficacy, and health consciousness on dietary behaviors. Second, perceived barriers are important for healthy eating intentions. Measures such as external incentive or motivational support should be used to reduce the barriers to individuals engaging in healthy eating and to reduce individuals’ concerns about engaging in healthy eating. Third, health consciousness is the most important factor affecting the healthy eating intentions. Government agencies must increase information and publicity on healthy and unhealthy diets. Fourth, for perceived barriers, self-efficacy and health consciousness, measures should be taken to have a dual impact on intentions and behaviors. For perceived susceptibility, perceived severity, and perceived benefits, measures should be taken to intervene in intentions, so as to avoid the unnecessary waste of resources.

### 5.3. Limitations

First, this paper used a Likert scale approach to inquire, relying on self-report rather than actual behavior, and participants may be reluctant to express their true views due to social expectations and moral pressures, so findings should be treated with caution. Second, the survey design used a cross-sectional approach, so it is only able to capture beliefs and behavioral intentions at a single point in time. Given that beliefs and behavioral intentions change over time, future research could explore this from a para-experimental perspective or use time-series data for follow-up studies. Third, since HBM is based on a rationally developed framework, irrational factors may be ignored. Therefore, in future studies, more theoretical models can be tested to consider the effects of different psychological constructs on eating intentions and behaviors.

## Figures and Tables

**Figure 1 ijerph-19-09037-f001:**
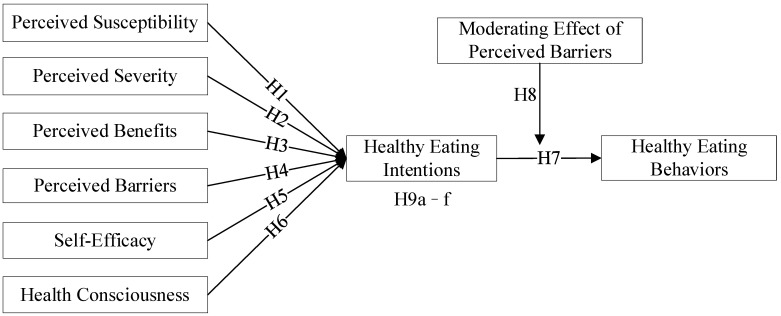
Theoretical hypothesis model.

**Figure 2 ijerph-19-09037-f002:**
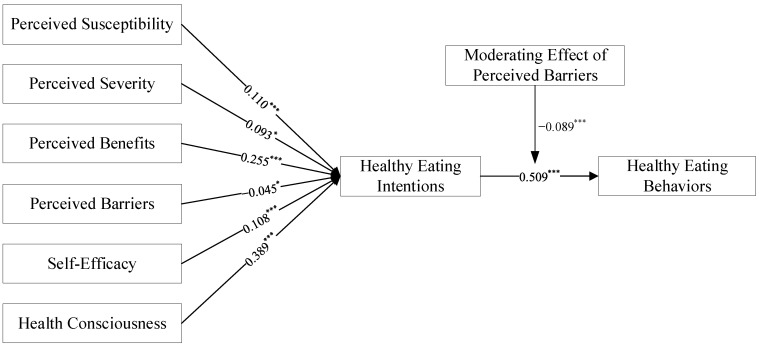
Structural model with path weight and significance level; Note: * and *** indicate significant at the 10%, and 1% significance levels, respectively.

**Figure 3 ijerph-19-09037-f003:**
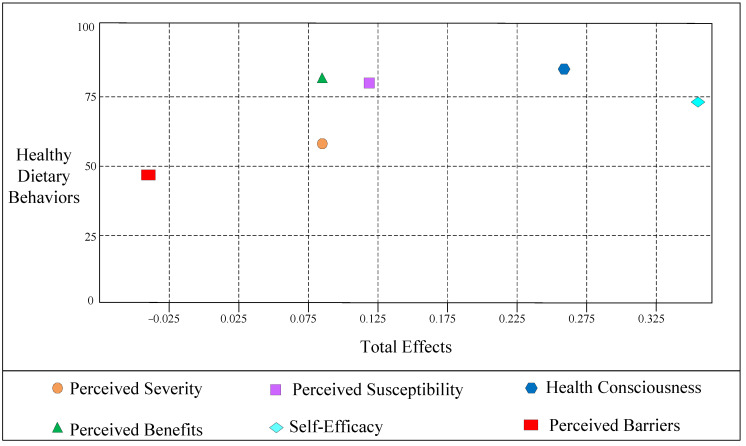
Importance performance map.

**Table 1 ijerph-19-09037-t001:** Demographic characteristics.

	Frequency (n)	Percentage (%)		Frequency (n)	Percentage (%)
Residence			Marital status		
Urban	924	72.13	Married	632	49.34
Rural	357	27.87	Unmarried	649	50.66
Gender			Body mass index		
Male	571	44.57	BMI < 18.5	156	12.18
Female	710	55.43	18.5 ≤ BMI < 24	754	58.86
Age			BMI ≥ 24	371	28.96
18~25 years old	462	36.07	Employment status		
26~30 years old	273	21.31	Attend school	216	16.86
31~40 years old	281	21.94	Employment	868	67.76
41~50 years old	168	13.11	Retirement	43	3.36
above 51 years old	97	7.57	Unemployed	154	12.02
Education			Family income (per year)		
Primary and below	20	1.56	USD < 15,780	646	50.43
Junior high school	114	8.90	USD 15,780~31,560	411	32.08
High school/secondary	198	15.46	USD 31,560~47,340	132	10.31
Junior College/Bachelor	811	63.31	USD 47,340~63,120	46	3.59
Graduate student	138	10.77	USD ≥ 63,120	46	3.59

Note: The range of annual household income includes the lower bound but not the upper bound.

**Table 2 ijerph-19-09037-t002:** Reliability and validity analysis.

Construct	Items	Loading	VIF	Cronbach’s Alpha	CR	AVE
Perceived Susceptibility (SUS)	SUS1	0.764	2.009	0.730	0.846	0.648
	SUS2	0.847				
	SUS3	0.801				
Perceived Severity (SEV)	SEV1	0.880	2.452	0.897	0.936	0.830
	SEV2	0.928				
	SEV3	0.924				
Perceived Benefits (BEN)	BEN1	0.879	2.057	0.871	0.921	0.795
	BEN2	0.901				
	BEN3	0.894				
Perceived Barriers (BAR)	BAR1	0.817	1.050	0.819	0.887	0.725
	BAR2	0.816				
	BAR3	0.918				
Self-Efficacy (SE)	SE1	0.859	1.469	0.857	0.913	0.777
	SE2	0.889				
	SE3	0.896				
Health Consciousness (HC)	HC1	0.866	1.748	0.839	0.903	0.757
	HC2	0.849				
	HC3	0.894				
Healthy Eating Intentions (HEI)	HEI1	0.804	1.000	0.887	0.922	0.749
	HEI2	0.866				
	HEI3	0.895				
	HEI4	0.894				
Healthy Eating Behaviors (HEB)	HEB1	0.732	-	0.835	0.879	0.547
	HEB2	0.751				
	HEB3	0.728				
	HEB4	0.718				
	HEB5	0.723				
	HEB6	0.785				

**Table 3 ijerph-19-09037-t003:** The correlation coefficients of latent variables and discriminant validity.

	SUS	SEV	BEN	BAR	SE	HC	HEI	HEB
*Fornell–Larcker Criterion*								
SUS	**0.805**							
SEV	0.689	**0.911**						
BEN	0.582	0.669	**0.892**					
BAR	−0.030	−0.041	−0.068	**0.851**				
SE	0.164	0.218	0.280	−0.218	**0.882**			
HC	0.357	0.438	0.477	−0.118	0.540	**0.870**		
HEI	0.480	0.535	0.600	−0.139	0.438	0.655	**0.865**	
HEB	0.233	0.281	0.316	−0.141	0.536	0.507	0.509	**0.740**
*Heterotrait–Monotrait Ratio (HTMT)*								
SUS	-							
SEV	0.843	-						
BEN	0.722	0.757	-					
BAR	0.073	0.059	0.075	-				
SE	0.198	0.248	0.323	0.237	-			
HC	0.449	0.506	0.558	0.122	0.634	-		
HEI	0.591	0.602	0.686	0.150	0.500	0.758	-	
HEB	0.283	0.317	0.363	0.144	0.635	0.600	0.581	-

Note: The diagonal (bold) elements are the square roots of AVEs and the off-diagonal elements are the correlations among constructs.

**Table 4 ijerph-19-09037-t004:** Results of hypothesis tests.

Hypo	Path	Beta	S.D.	*p*-Value	Confidence Interval	*f* ^2^	*R* ^2^	*Q* ^2^	Decision
H1	SUS -> HEI	0.110 ***	0.027	0.000	[0.056, 0.161]	0.151	0.563	0.419	Support
H2	SEV -> HEI	0.093 *	0.038	0.013	[0.023, 0.170]	0.212			Support
H3	BEN -> HEI	0.255 ***	0.041	0.000	[0.176, 0.336]	0.173			Support
H4	BAR -> HEI	−0.045 *	0.019	0.018	[−0.083, −0.008]	0.204			Support
H5	SE -> HEI	0.108 ***	0.026	0.000	[0.058, 0.159]	0.168			Support
H6	HC -> HEI	0.389 ***	0.034	0.000	[0.323, 0.453]	0.199			Support
H7	HEI -> HEB	0.509 ***	0.025	0.000	[0.457, 0.554]	0.351	0.358	0.237	Support
Moderating Effect of Perceived Barriers									
H8	Interaction item -> HEB	−0.089 ***	0.021	0.000	[−0.120, −0.042]	-			Support

Note: * and *** indicate significant at the 10%, and 1% significance levels, respectively.

**Table 5 ijerph-19-09037-t005:** Mediating effects.

Hypo	Associations	Direct Effects	Indirect Effects	Total Effects	VAF	Decision
H9a	SUS -> HEI -> HEB	0.014 (0.489)	0.248 *** (11.023)	0.262 *** (7.291)	0.947	Support
H9b	SEV -> HEI -> HEB	0.012 (0.417)	0.269 *** (11.640)	0.281 *** (8.754)	0.957	Support
H9c	BEN -> HEI -> HEB	0.017 (0.579)	0.300 *** (12.437)	0.316 *** (10.760)	0.949	Support
H9d	BAR -> HEI -> HEB	−0.084 ** (3.267)	−0.071 ** (5.290)	−0.155 *** (5.704)	0.458	Support
H9e	SE -> HEI -> HEB	0.398 *** (13.471)	0.148 *** (9.334)	0.546 *** (22.841)	0.271	Support
H9f	HC -> HEI -> HEB	0.313 *** (9.359)	0.199 *** (8.581)	0.512 *** (20.803)	0.389	Support

Note: **, and *** indicate significant at the 5%, and 1% significance levels, respectively; VAF: variance accounted for; values in parentheses are T values.

**Table 6 ijerph-19-09037-t006:** Multi-group analysis.

Path	Gender	Age	Education	Income	BMI	Residence	Marital Status	Employment Status
SUS -> HEI	0.285	0.227	0.883	0.052	0.656	0.174	0.980	0.834
SEV -> HEI	0.338	0.183	0.769	0.500	0.844	0.265	0.533	0.179
BEN -> HEI	0.515	0.295	0.505	0.368	0.527	0.669	0.557	0.630
BAR -> HEI	0.170	0.783	0.896	0.772	0.895	0.341	0.197	0.289
SE -> HEI	0.857	0.216	0.219	0.319	0.931	0.708	0.700	0.125
HC -> HEI	0.813	0.501	0.733	0.065	0.221	0.495	0.470	0.279
HEI -> HEB	0.655	0.417	0.049	0.928	0.170	0.958	0.291	0.538
SUS -> HEI -> HEB	0.183	0.385	0.532	0.398	0.117	0.538	0.581	0.864
SEV -> HEI -> HEB	0.562	0.156	0.865	0.416	0.512	0.227	0.283	0.544
BEN -> HEI -> HEB	0.871	0.627	0.191	0.750	0.331	0.573	0.614	0.940
BAR -> HEI -> HEB	0.111	0.781	0.836	0.507	0.492	0.750	0.508	0.759
SE -> HEI -> HEB	0.810	0.156	0.067	0.597	0.281	0.968	0.221	0.132
HC -> HEI -> HEB	0.363	0.517	0.328	0.276	0.140	0.956	0.121	0.408
Interaction item -> HEB	0.053	0.292	0.074	0.347	0.404	0.417	0.424	0.586

Note: The values in the table are *p*-values.

**Table 7 ijerph-19-09037-t007:** Data of the importance performance map for healthy dietary behaviors.

Associations	Total Effect	Performance
Perceived Susceptibility	0.112	83.706
Perceived Severity	0.085	64.680
Perceived Benefits	0.086	83.998
Perceived Barriers	−0.030	48.228
Self-Efficacy	0.352	74.046
Health Consciousness	0.258	84.421

## Data Availability

The data presented in this study are available on request from the corresponding author. The data are not publicly available due to privacy restrictions.

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
