# Peer review of "Chinese Residents’ Healthy Eating Intentions and Behaviors: Based on an Extended Health Belief Model"

_ijerph, 2022, doi:10.3390/ijerph19159037_

Round 1

Reviewer 1 Report

1.    The authors start from the premise that internet users are young and educated, therefore, their diet is healthier. I think that's a false premise. On the contrary, these young people who spend a lot of time in front of the computer tend to feed mainly on fast food, associated with carbonated drinks. On the other hand, people from rural areas embrace traditional Chinese food based on cereals, fruits and vegetables. In addition, the people of the Chinese country also make sustained physical effort. I ask the authors to clarify this contradiction. The famous studies of the nutritionist Thomas Campbell (The Chinas Study, Integral, etc.) show that rural China is much healthier than urban China precisely because of its diet and lifestyle.

2.       It would be interesting to know the attitude of the respondents towards nutritional supplements. In general, people find it very difficult to change their diet, but they are willing to compensate for this refusal with the consumption of nutritional supplements. That is, they respond exactly to the interests of the pharmaceutical industry and the manufacturer of nutritional supplements.

3.       In Chapter 4 (results and discussions), the paper should be improved by comparing it with the results of other similar studies. From this point of view, the work is quite poor. Almost all the comments revolve around the results of the research in this paper.

4.       In order to broaden the scope of the readers of this article, I suggest the authors to express some conclusions related to the study, without integrating everything into the logic and terminology of statistical calculation.

The topic of the paper is of major interest, given the increasingly poor health of the human population. The work is very well structured and generally meets most of the scientific rigors. From my point of view it can be published after some improvements.

Reviewer 2 Report

This study investigated the association between healthy eating intentions and behaviors, and also examined the factors of healthy eating, which was a interesting question. The sample was large, and the results are reliable. I have some suggestions to improve the quality of paper.

1. BMI was a important variable in eating behaviors, did you measure it and control it in data analysis?

2. The scales that are used in this study are most in self-made measures, and the relationship between four variables are high in correlations. I think, it will influence the true relationship between them.

3.  CFA and EFA should be conducted in different samples.

4.  Did you make some improvements in HBM? Or just replicated it?

Reviewer 3 Report

IJERPH #1795131

Title: “Factors Influencing Healthy Eating”

Major Concerns:

I see several statistical tests being used to reveal correlations between survey subsections, but these are NOT really “factors” that affect diets and eating behavior. All I see are tests to show how some survey answers are related to other answers. If you are not prepared to do additional research on food consumption, you should change the title of this article. The title you have now is misleading. There are also serious problems with the wording of the survey Likert items.

Appendix A: Survey Items

First, the content of the survey and the exact wording of the survey statements needs to be presented in the methods section. The survey items are incomplete and this weakens the validity of the study and the data. Several important concepts are not defined including “healthy diet,” “unhealthy diet,” “healthy eating,” and eating “fairly lightly.” Since these phrases and concepts are NOT clearly defined by the authors, how do you know what the research subjects think these mean when they agree or disagree with the statements? If you do not define “healthy eating,” then how can the research subjects agree or disagree, to know if they are doing this? 

Abstract: Need to Revise:

Line 12: One major claim is not clear and not supported by fact. The authors say that “unhealthy eating” is the “primary” cause of “the current” health problems in China. But, what are the health problems? Do the authors mean obesity, heart disease, or some types of cancer? It is not clear.

Line 16: They say that “1281 data” were obtained. This is also not clear. It would be more accurate to say that data were collected from 1281 completed surveys.

Summary Is Misleading:

Lines 18-27: The summary shows that this is an incomplete study, or that the title is misleading, or both. The methods and results show that several correlations were discovered between the subsections of survey, and that several statistical tests were performed on the Likert-items, but this study does not identify “factors” actually influence diet and healthy eating in real life.   

Body: Needs Editing:

Lines 32-33: Not clear what the authors mean by a “change in structure” of diet. Do you mean composition of meals or sources and quantities of food?

Line 43: Authors say that “obesity” is a health problem, but do not identify what the “other” food-related problems are in China. More data are needed.

Line 55: Not clear how evidence from Bettina (5) is used. Are you saying that people with diabetes are not likely or less likely to change their diets, compared to the general population?

Sections 2.1 to 2.7: Hypotheses

Lines 100-170: There are too many hypotheses. This shows a lack of focus. This looks like an exercise in discovering as many correlations in data as possible, based on p-values, but not a study of the actual social and behavioral factors that really influence healthy eating. The “background” section is also incomplete. It reviews the theory of the HBM model, but there is not any relevant data on diet and eating behavior in China.

Survey Methods:

Line 184: If the average time to complete the survey questions was only 7 minutes, this suggests that research subjects did not spend much time considering their answers. This also does not sound like a rigorous investigation of eating behavior and diet, that would require more time.

Line 180: The authors say that subjects “promised” to tell the truth, but if this was only done on-line, there is no way to know if they did tell the truth. There is no guarantee, and by not varying the type of questions and survey items, there is no way to reinforce consistent and truthful answers. I see several statistical tests being used to reveal correlations between survey subsections, but these are NOT really “factors” that affect diets and eating behavior. All I see are tests to show how some survey answers are related to other answers. If you are not prepared to do additional research on food consumption, you should change the title of this article. The title you have now is misleading. There are also serious problems with the wording of the survey Likert items.

Lines 317-318: The authors say that “raising health consciousness” is directly related to intentions to eat more healthy,” but say nothing about how “consciousness” can be raised effectively. This should be part of the Background section. There should be some review of successful efforts to teach nutrition or health education in China or other attempts to improve eating habits.

Section 5.2

Lines 407-417: These suggestions are not adding anything new to the discussion of the HBM model or to the previous analyses and critiques. It would be much more helpful to know more successful or unsuccessful attempts to improve diet and eating in China. More discussion of the statistical methods is not helpful, based on the title of this article.

Section 5.3: Limitations

Lines 419-428: The Likert-scale is NOT the problem. The bigger problem is assuming that answers are honest and accurate and also the data that comes from this survey. The subjects only spent an average of 7 minutes on the survey questions and there are also problems with the wording and the way that the survey statements are written.

Appendix A: Survey Items and Wording

Line 445: First, the content of the survey and the exact wording of the survey statements needs to be presented in the methods section. The survey items are incomplete and this weakens the validity of the study and the data. Several important concepts are not defined including “healthy diet,” “unhealthy diet,” “healthy eating,” and eating “fairly lightly.” Since these phrases and concepts are NOT clearly defined by the authors, how do you know what the research subjects think these mean when they agree or disagree with the statements? If you do not define “healthy eating,” then how can the research subjects agree or disagree, to know if they are doing this?  

Round 2

Reviewer 2 Report

The authors have answered the questions well, and I have no new suggestions. I think, it can be accepted to publish in the journal.